# Antibacterial Activity of Different Blossom Honeys: New Findings

**DOI:** 10.3390/molecules24081573

**Published:** 2019-04-21

**Authors:** Marcela Bucekova, Lucia Jardekova, Valeria Juricova, Veronika Bugarova, Gabriele Di Marco, Angelo Gismondi, Donatella Leonardi, Jarmila Farkasovska, Jana Godocikova, Maros Laho, Jaroslav Klaudiny, Viktor Majtan, Antonella Canini, Juraj Majtan

**Affiliations:** 1Laboratory of Apidology and Apitherapy, Department of Microbial Genetics, Institute of Molecular Biology, Slovak Academy of Sciences, Dubravska cesta 21, 845 51 Bratislava, Slovakia; marcela.bucekova@gmail.com (M.B.); jardekovalucia@gmail.com (L.J.); valeria.juricova@gmail.com (V.J.); bugarova9@gmail.com (V.B.); jarmila.farkasovska@savba.sk (J.F.); jana.godocikova@savba.sk (J.G.); maros.laho@savba.sk (M.L.); 2Honey Research Center, Department of Biology, University of Rome “Tor Vergata”, via della Ricerca Scientifica 1, 00133 Rome, Italy; gabriele.di.marco@uniroma2.it (G.D.M.); gismondi@scienze.uniroma2.it (A.G.); leonardi@uniroma2.it (D.L.); canini@uniroma2.it (A.C.); 3Institute of Chemistry, Slovak Academy of Sciences, Dubravska cesta 9, 845 38 Bratislava, Slovakia; jaroslav.klaudiny@savba.sk; 4Department of Microbiology, Faculty of Medicine, Slovak Medical University, Limbova 12, 833 03 Bratislava, Slovakia; viktor.majtan@szu.sk

**Keywords:** glucose oxidase, hydrogen peroxide, polyphenols, mechanism of action

## Abstract

Antibacterial activity is the most investigated biological property of honey. The goal of this study was to evaluate the antibacterial activity of 57 Slovak blossom honeys against *Staphylococcus aureus* and *Pseudomonas aeruginosa* and investigate the role of several bioactive substances in antibacterial action of honeys. Inhibitory and bactericidal activities of honeys were studied to determine the minimum inhibitory and bactericidal concentrations. The contents of glucose oxidase (GOX) enzyme, hydrogen peroxide (H_2_O_2_), and total polyphenols (TP) were determined in honeys. We found that honey samples showed different antibacterial efficacy against the tested bacteria as follows: wildflower honeys > acacia honeys > rapeseed honeys. Overall antibacterial activity of the honeys was statistically-significantly correlated with the contents of H_2_O_2_ and TP in honeys. A strong correlation was found between the H_2_O_2_ and TP content. On the other hand, no correlation was found between the content of GOX and level of H_2_O_2_. Antibacterial activity of 12 selected honeys was markedly reduced by treatment with catalase, but it remained relatively stable after inactivation of GOX with proteinase-K digestion. Obtained results suggest that the antibacterial activity of blossom honeys is mainly mediated by H_2_O_2_ levels present in honeys which are affected mainly by polyphenolic substances and not directly by GOX content.

## 1. Introduction

Honey, a bee product, offers a broad-spectrum of biological properties including antimicrobial and wound healing activity [1]. Importantly, antibacterial and anti-biofilm potentials of honey have been considered as an exclusive criterion of its wound healing properties. Therefore, the antibacterial activity of this matrix from different floral sources has been intensively studied over the past few decades. It has been found that the botanical origin of honey influences it, due to its content of phytochemicals [2,3,4]. Some of these potent natural honeys are currently being used as medical-grade substances in dermatological and other medical practices under different brands, e.g. Medihoney (manuka-based honey), Revamil (blossom-based honey), and Honevo (kanuka-based honey) [5,6,7].

The mechanism underlying the antibacterial activity of honey has not been fully elucidated. Besides high sugar content, low pH, and water activity, it has been considered that hydrogen peroxide (H_2_O_2_) and the bee-derived antibacterial peptide defensin-1 (Def-1) are the major components responsible for the antibacterial activity of honey, with the exception of manuka honey [8]. Very recent studies [9,10,11,12,13] described that the antibacterial effect of European honeys was mostly due to the presence of H_2_O_2_. The levels of H_2_O_2_ may differ from honey to honey, and several factors may affect the total concentration of H_2_O_2_ [14]. It is thought that H_2_O_2_ is generally generated by glucose oxidase (GOX)-mediated conversion of glucose to gluconic acid under aerobic conditions in diluted honey [15]. On the other hand, we recently reported that H_2_O_2_ in diluted honeydew honeys is likely produced via an alternative non-enzymatic pathway, where plant polyphenols may take part in the process of gradual H_2_O_2_ production [10]. Although, H_2_O_2_ is considered to be a key antibacterial compound in diluted honey, some studies showed that its level in different honeys did not correlate with their overall antibacterial activity [10,16]. More importantly, the H_2_O_2_ levels seem to be the ability of honeys to generate hydroxyl radicals from degraded H_2_O_2_ via Fenton reaction. Brudzynski and Lannigan (2012) reported that oxidative stress caused by honey action on bacterial cells resulted from hydroxyl radical generated from honey’s H_2_O_2_. It was also reported that H_2_O_2_ in aqueous solution did not oxidize nucleic acids, lipids, or proteins in absence of transition metal ions such as Cu(I) or Fe(II) [17]. In addition, H_2_O_2_ hydrolysis also produces oxygen, which can speed up the auto-oxidation of honey polyphenols, which, upon becoming pro-oxidant agents generate further H_2_O_2_ molecules and drive the generation of hydroxyl radicals from H_2_O_2_ in the presence of transition metals [18]. Thus, the plant-derived polyphenolic compounds that are always present in honeys can significantly contribute to and/or modulate the antibacterial effects of honey.

The aim of the present study was to (i) determine the antibacterial activity of various blossom types of Slovak honey samples against two wound pathogens, *Staphylococcus aureus* and *Pseudomonas aeruginosa*; (ii) determine the contents of GOX, total polyphenols, and H_2_O_2_ in the analyzed honeys; and (iii) investigate the role of H_2_O_2_ in blossom honey antibacterial activity.

## 2. Results

### 2.1. Characterization of Pollen Profile in Different Honey Types

#### 2.1.1. Acacia Honeys

Melissopalynological analysis of acacia samples (Figure 1A) documented the presence of *R. pseudoacacia* pollen with a frequency of 15–45% in all samples, as expected for a hypo-represented monofloral honey. Moreover, in these samples, *Rubus sp*. and Brassicaceae pollens were also detected with a frequency of <3% in 62.5 and 12.5% of the honeys, respectively; a frequency between 3 and 15% in 25% of the samples (for both pollen types); a frequency between 15 and 45% in 12.5 and 50.0% of the samples; and for only Brassicaceae, a frequency of >45% in 12.5% of the honeys. Nectariferous taxa with early flowering, for example *Salix* sp. and *Prunus* sp. (respectively present in 75.0 and 62.5% of the samples with variable percentage), *Acer* sp. and fruit bearing species (e.g., *Malus/Pyrus* sp.) were identifiable in this class of honeys. In particular, pollen granules, typically associated to Alpine area, like *Vicia* sp. and *Loranthus* sp., were also found in *R. pseudoacacia* honeys, as documented by Persano Oddo et al. [19].

#### 2.1.2. Wildflower Honeys

Pollen analysis of these samples (Figure 1B) showed the presence of *Rubus* sp. (with a frequency between 15 and 45% in 23.7% of the samples, between 3 and 15% in 57.9%, and <3% in 15.8%) and Brassicaceae pollen (with a frequency >45% in 31.6% of the samples; and between 15–45%, between 3–15%, and <3% in 21.0, 10.5, and 23.7% of the honeys, respectively). Furthermore, the presence of pollen of various early flowering taxa was identified in these honeys such as *Trifolium* sp. (detected in 23.7% of the honeys with a frequency <3%, in 42.1% between 3–15%, in 7.9% between 15–45%, and >45% in 2.7%); *Salix* sp. (in 5.3% of the samples with a frequency >45%, in 23.7% between 15–45%, in 34.2% between 3–15% and in 23.7% with <3%); *Acer* sp. (present in 10.5% of the honeys with a frequency <3%, in 31.6% between 3–15%, in 13.2% between 15–45% and in 2.7% with >45%); Apiaceae (identified in 2.7% of the samples between 15 and 45%, in 21% between 3 and 15% and in 34.2% with <3%); *Prunus* sp. (present in 55.2% of the honeys; with a frequency between 3–15% and <3% in 36.8% and 18.4% of the samples, respectively); Compositae C, H, and T types (found in 2.6% of the samples with a frequency >45%; and between 3–15% and <3% in 18.4 and 39.5%, respcetively); *R. pseudoacacia* (detected in 2.6% of the honeys with a frequency between 15–45%, while at a frequency of 3–15% and <3% in 31.6 and 5.3% of the honeys, respectively); *Acer* sp. (revealed in 2.6% of the samples with a frequency >45% in 13.2% between 15 and 45%, in 31.6% between 3 and 15% and in 10.5% with <3%); and Poaceae (in 39.5% of the samples with a frequency <3%).

#### 2.1.3. Rapeseed Honeys

These samples (Figure 1C) showed the presence of Brassicaceae pollen with a frequency of always >45% as expected, since it is dominant and hyper-represented in this plant species. Consequently, these rapeseed honeys appeared poor of other *taxa*, which could be detected only in traces with a frequency of <20%. We observed the presence of early flowering plants, such as *Myosotis* sp., (in 12.5% of samples with a percentage of <20%), *Salix* sp. (in 25% of the honeys with a frequency between 3–15%, and in 75% with <3%), *Prunus* sp. (detected in 50% of the samples with a frequency between 3–15%, and in 37.5% with <3%), *Rubus* sp. (found in 25% of the samples with a frequency between 3–15%, and in 62.5% with <3%), *Malus*/*Pyrus* sp. (present in 12.5% of the samples with a frequency between 3–15% and in 50% with a frequency <3%) and *Acer* sp. (present in 87.5% of the samples with a frequency <3%).

#### 2.1.4. Raspberry and Phacelia Honeys

*Rubus idaeus* and *Phacelia* sp. pollen are present in the respective samples with a frequency of >45%, coherently with the definition of monofloral honeys (Figure 1D). *Rubus idaeus* and *Phacelia* sp. are plant species typical of the Eastern and Northern Europe [19]. Characteristic spring-like *taxa* were also found: *Trifolium repens* (in the samples with a mean frequency <15%), *Myosotis* sp. (in raspberry honeys with a frequency between 3–15%), *Vicia* sp. and *Prunus* sp. (both present in the samples with a frequency of <3%).

### 2.2. Antibacterial Activity of Blossom Honey Samples

The minimal inhibitory concentration (MIC) and minimal bactericidal concentration (MBC) values of 57 blossom honey samples against two tested bacterial species are shown in Figure 2. For *S. aureus*, the MIC values of the honey samples ranged from 6.0 to 45.0% (concentrations of honeys in solution), whereas the MICs of honeys against *P. aeruginosa* ranged from 8.0 to 40.0%. Wildflower honeys (W16–W47), together with raspberry (Ra48) and phacelia (P49) honeys, exhibited the strongest antibacterial activities giving MIC average values of 13.9 and 15.0% against *S. aureus* and *P. aeruginosa*, respectively. On the other hand, acacia (A1–A15) and rapeseed honeys (R50–R57) exhibited weaker antibacterial activities, giving an average MIC values between 18.7–19.1% versus of 21.2–22.5% against *S. aureus* and *P. aeruginosa*, respectively (Figure 3). Overall, wildflower honeys showed statistically-significantly higher antibacterial efficacy in comparison to acacia and rapeseed honeys (Figure 3).

All honey types contained samples showing low antibacterial activity (MIC values lower than 20%). The MIC values of three honey samples (W19, W42, and R56) were above 30%, suggesting that these samples were most likely overheated or stored under unsuitable conditions. Six samples (W16, W22, W26, W27, W43, and W44) were the most effective against both tested bacteria, giving average values below 10%.

The profile of MBC values was similar to that of MIC values (Figure 2). In particular, MBC values were either identical or slightly higher (one dilution above) than MIC values.

Manuka honey exhibited comparable or stronger antibacterial activity than blossom Slovak honey samples. Manuka honey was, similarly to blossom honeys, more effective against *S. aureus* than *P. aeruginosa*.

### 2.3. GOX Content and Antibacterial Activity of Honey Samples

The method for quantification GOX in honeys by the wet immunoblot assay based on prepared recombinant GOX (rGOX) and rabbit polyclonal anti-honeybee GOX antibody was developed. Both specificity and affinity of the antibody used was previously determined [20]. No cross-reaction with other honey proteins separated by SDS-PAGE could be detected. Prepared rGOX in denaturing solution (4 M urea, 20 mM Tris pH 8.0) had 70% purity (determined by SDS-PAGE). A strong linear relationship (r = 0.984) between rGOX amount and band volume intensity was observed in the calibration curve created for GOX quantification in honeys (Figure 4).

The GOX content in analyzed blossom honey samples, determined by the established assay, ranged from 5.0 to 56.4 μg/g with an average value of 29.7 μg/g. Acacia honey samples (A1–A15) contained the highest amount of GOX, with an average value of 33.7 μg/g of honey (Figure 5A). The lowest content, with average value of 24.3 μg/g of honey, was determined in rapeseed samples.

No correlation was found between GOX content and MIC values (Figure 2) for both tested bacteria, among the blossom honey samples (r = 0.022, *P* = 0.87 for *S. aureus* and r = 0.001, *P* = 0.99 for *P. aeruginosa*). This observation suggested that the level of GOX in honey was not a critical factor for overall honey antibacterial activity.

### 2.4. H_2_O_2_ and Total Polyphenol Contents in Honey Samples

The content of H_2_O_2_ in blossom honey samples was determined in 40% honey solutions (pH 7.0) immediately after preparation of homogenous solutions. The content ranged from 2.4 to 47.2 μg/g of honey (Figure 5B). The average value of H_2_O_2_ content was 13.5 μg/g of honey. Polyfloral honeys (W16–W47) showed the greatest amount of H_2_O_2_, with an average value of 18.1 μg/g of honey with respect to acacia and rapeseed honeys which had average values of 6.8 and 7.8 μg/g of honey, respectively. Correlation test showed no significant correlation between the values of H_2_O_2_ and GOX content among the honey samples (r = −0.087, *P* = 0.521). These results suggest that there is no direct relationship between the level of GOX and the level of H_2_O_2_ in blossom honeys.

One of the important groups of biologically active molecules in honey are polyphenols/flavonoids. It is considered that these molecules can elevate the levels of H_2_O_2_ and thus improve the antibacterial activity of honey. TP content in blossom honey samples was determined in the 20% (*w*/*v*) honey solutions and it is shown in Figure 5C. The content in honey samples ranged from 91 to 452 μg GAE/g of honey with an average value of 228 μg GAE/g of honey. Acacia honey samples (A1–A15) showed the lowest content, with an average value of 126 μg GAE/g of honey (Figure 5C). The highest content, with an average value of 274 μg GAE/g of honey, was determined, as expected, in wildflowers honey samples (W16–W47). High TP contents were also found in raspberry and phacelia honeys.

In contrast to results from statistical analysis between the GOX and H_2_O_2_ contents, a strong statistically-significant correlation (r = 0.66, *P* < 0.0001) between the TP and H_2_O_2_ content was found in blossom honey samples.

### 2.5. Role of Honey Proteins and H_2_O_2_ in the Antibacterial Effect of Blossom Honeys

Our recent findings showed that bee-derived protein components, including GOX and Def-1, are not responsible for the pronounced antibacterial activity of the honeydews honey [10]. Therefore, we investigated the contribution of the proteins in selected blossom honeys showing the strong antibacterial activity (four acacia (A1, A2, A8, and A9), four wildflower (W21, W22, W26, and W27), and four rapeseed (R52, R53, R55, and R57) honeys) to the overall honey antibacterial effect. Upon proteinase K treatment of 50% (*w*/*w*) honey solutions, diluted with Mueller–Hinton broth (MHB), proteins including GOX were digested (Figure 6). This digestion of GOX in honey solution resulted in a loss of its enzymatic activity. We indirectly verified the loss of activity using a partially purified native GOX from honey. Its enzymatic activity was completely abolished with proteinase-K treatment performed in the same conditions as in the case of honey solution (data not shown).

After confirming the proteinase K inactivation effect on GOX, antibacterial activities of proteinase K-untreated and -treated honey samples were studied. Two kinds of untreated samples were used; non-incubated and 2-h-incubated (catalase treatment time) at room temperature after honey dilution in MHB. Differences in antibacterial activity were observed between non-incubated and 2-h-incubated untreated honeys depending on certain types of honey. In all acacia honey samples, the 2-h incubation caused increases of MIC values in comparison with non-incubated samples. In contrast, 2-h incubation led to decrease of MIC values in all wildflower and most rapeseed honeys. The changes were most pronounced in the case of *S. aureus* than *P. aeruginosa* (Figure 7). This finding suggests a significant catalase action and/or weaker H_2_O_2_ production capacity in diluted acacia honeys (having high GOX and low TP content) than it was in other two types of honey. No changes (*S. aureus*) or weak increasing of MIC values (*P. aeruginosa,* except A1 honey sample) were observed between the untreated 2-h-incubated and proteinase K-treated honey samples. The last results suggest that action of protein components GOX and Def-1 in diluted honeys do not markedly contribute to the antibacterial activity of the blossom honeys.

The above findings supported the idea that H_2_O_2_, accumulated during the ripening of honey, has a major role in antibacterial activity of blossom honeys. It seems that the further production of H_2_O_2_ in diluted honeys incubated at room temperature for 2 h contributed only partially to honey antibacterial activity. Moreover, production of H_2_O_2_ was not connected with GOX action.

To demonstrate the role of H_2_O_2_ in antibacterial activity of honey samples, a catalase treatment of honeys was performed. Catalase treated honeys had significantly lower (*P* < 0.001) antibacterial activity, with MIC values of 25–42% and 25–40% against *S. aureus* and *P. aeruginosa*, respectively, compared to non-treated samples having MIC values 3–27% and 8–25%, respectively. As expected, the MIC values of manuka honey following catalase or proteinase-K treatment did not change (Figure 7).

The obtained results proved that H_2_O_2_ was major antibacterial factor of blossom honeys. This finding is also supported by a statistical analysis of the relationship between the level of total H_2_O_2_ and overall antibacterial activity of blossom honeys. Significant correlations between these two parameters were found in case of *S. aureus* (r = −0.48, *P* < 0.001) and *P. aeruginosa* (r = −0.65, *P* < 0.0001) (Figure 8A,B). However, in spite of the mentioned overall correlations, there are a number of honey samples at which a weak correlation or no correlation can be observed. These honeys showed either similar MICs, 7–10%, at different concentrations of H_2_O_2_, 15–31 μg/g of honey; or markedly different MIC, 11–43%, at low concentrations of H_2_O_2_, 3.5–9 μg/g of honey (Figure 8A).

To provide evidence of direct antibacterial activity of H_2_O_2_ against tested bacterial strains, we determined the MICs of artificial honey (AH) and AH enriched with H_2_O_2_ (at concentration of 50 μg/g of AH). A significant decrease of MIC values of AH enriched with H_2_O_2_ was demonstrated (Figure 9). This result suggests that H_2_O_2_ accumulated during the ripening of blossom honey can itself (not through the substance(s) derived by action of H_2_O_2_ and some other honey components) significantly affect the overall honey antibacterial activity.

### 2.6. Role of Polyphenols in the Antibacterial Effect of Blossom Honeys

Statistical analysis revealed a significant correlation between the TP content and overall antibacterial activity of blossom honeys against *S. aureus* (r = −0.40, *P* < 0.01) and *P. aeruginosa* (r = −0.47, *P* < 0.01) (Figure 8C,D). Nevertheless, similarly as it is in the case of H_2_O_2_, there are honey samples at which weak or no correlation exists among these parameters. Taking all obtained results in the work, it seems that the concentration of polyphenols in blossom honeys together with the level of H_2_O_2_ are critical factors in determining the honey antibacterial potential.

## 3. Discussion

Antibacterial activity is the most evaluated and investigated biological activity of honey. Nevertheless, the mechanism of honey antibacterial action has not been elucidated in a sufficient manner. In this study, we characterized the inhibitory and bactericidal activity of 57 Slovak blossom honeys; determined contents of GOX (by an established a semi-quantitative assay), H_2_O_2_, and TP in honeys; and evaluated the role of these components in antibacterial activity.

Numerous studies have examined the antibacterial activity of honey from different botanical and geographical origin. Some types of honeys (e.g., buckwheat, fir honeydew honey) were more potent than others, but all honeys, except manuka honey, contained the same antibacterial substance, H_2_O_2_. Currently, it is believed that H_2_O_2_ is mainly produced by GOX-mediated conversion of glucose to gluconic acid under aerobic conditions in diluted honey [15]. However, the results presented here, together with our previous recent observations [10]—as well as results from other studies [21,22,23]—support the theory by which H_2_O_2_ in honey is also produced via an alternative non-enzymatic pathway. Two facts were revealed in this study, (1) absence of a correlation between the contents of GOX and H_2_O_2_ occurring in blossom honeys, and (2) determination of the similar MIC values in diluted honeys not incubated and incubated with proteinase-K (destroying activity of GOX and obviously also other proteinous substances) provide evidence that GOX as well as Def-1 have no significant effect on the antibacterial activity of undiluted and diluted blossom honeys. Similar results were found in our recent study [10] where 23 honeydew honeys were analyzed and overall antibacterial activity was evaluated. In the recent study, statistical analysis did not show any correlation between antibacterial activity of honeydew honeys and level of H_2_O_2_. In contrast to these findings, the significant statistical correlation between these two parameters was observed within blossom honeys investigated in this work. A stronger correlation between MIC values against *P. aeruginosa* and H_2_O_2_ content than against *S. aureus* was found. Nevertheless, some of the examined blossom honey samples did not show any correlation among MICs and H_2_O_2_ content_._ We suppose that in these samples, the other factors associated with production of hydroxyl radicals and with variations in Cu(I) and Fe(II), polyphenols and catalase levels and other yet unknown factors affected their final antibacterial efficacy. Similarly, Grecka and co-workers (2018) determined the overall antibacterial activity and concentration of H_2_O_2_ in 144 honey samples from Northern Poland [12]. Although a significant correlation between the level of accumulated H_2_O_2_ and antibacterial activity of Polish honey samples was also documented, the samples with low MIC values (the most potent samples) were characterized with low concentration of H_2_O_2_. In contrast to our study, they determined H_2_O_2_ concentrations in 25% (*w*/*v*) honey solutions in deionized water after 1 h of incubation at 37 °C. Even though the different methodological approach in H_2_O_2_ determination resulted in statistically-significant correlations in both studies, we consider that the deeper understanding the role of H_2_O_2_ in honey antibacterial action will require a study of the level of generated H_2_O_2_ levels at individual MIC values of each honey.

One of the important observations in this study is that the initial content of H_2_O_2_ in undiluted blossom honey is able to inhibit the growth of both tested bacteria. Despite some published data reporting that cell death of bacteria requires concentrations of H_2_O_2_ higher than 50 mM [14] and the content of H_2_O_2_ in honey is 900-fold lower than the H_2_O_2_ content used in medical disinfectants [24], we showed here that H_2_O_2_ at concentration about 1.0 mM (ca. 50 μg/g of honey) in undiluted AH was able to inhibit bacterial growth which was documented by a decrease in the MIC values of artificial honey from 45 to 14% for *S. aureus* and from 25 to 17% for *P. aeruginosa*.

Depending on several factors but mainly on its concentration, H_2_O_2_ may exert either bacteriostatic or bactericidal effects. H_2_O_2_ at low concentrations, that are more stable than higher concentrations, exerts weak antibacterial activity; however, extended slow-release of H_2_O_2_ may improve its antibacterial activity and overwhelm the effect of bacterial/plant catalase [25].

Questions as to how H_2_O_2_ is generated in diluted honeys and other factors that affect H_2_O_2_ antibacterial action remain unanswered. We assumed that plant-derived polyphenolic compounds, often reported in honey samples, might be one factor responsible for H_2_O_2_ production and elevating honey antibacterial activity. Indeed, as shown in this study, a significant correlation between the concentration of TP and antibacterial activity of tested blossom honey was documented. These findings are in agreement with another study, where a strong correlation between the antibacterial activity of various honey types and phenolic content was documented [26]. Another critical factor affecting the content of H_2_O_2_ in honey can be presence of pollen-derived catalase, a plant enzyme responsible for degradation of H_2_O_2_. Interestingly, catalase has occasionally been reported in honey samples and the most proteomic studies have not been able to identify this enzyme in honey samples [27,28,29,30,31]. A more objective approach seems to be the identification of catalase in honey by determining its enzymatic activity [32]. Nowadays, it is difficult to evaluate the content of catalase in particular honey types because a reliable quantitative method is missing. In this study, we observed a strong catalase activity destroying H_2_O_2_ in the case of acacia honeys. The sharp association between the effect and the type of honey is interesting and will require further study.

Polyphenols, mostly flavonoids and phenolic acids that are used as markers of honey botanical origin and for its authentication [33], attract honey consumers’ interest because of their health-promoting and biological effects. The concentration of polyphenols and flavonoids in honey varies between 56 and 500 mg/kg and 0.6 and 6.4 mg/kg of honey, respectively [34,35,36,37,38]. A direct antibacterial effect of polyphenols in honey is negligible due to their very low concentration in honey. In fact, they may work in two ways to promote antimicrobial activity: 1) by producing H_2_O_2_ and 2) by reducing Fe (III) to Fe (II), which triggers the Fenton reaction and lead to creation of more potent reactive oxygen species such as hydroxyl radicals. Recently, Grzesik and co-workers [39] tested 54 natural antioxidants, including polyphenols, and revealed that 27 of them were able to generate H_2_O_2_ by autoxidation of polyphenols. No correlation between the structure of the studied compounds and generation of H_2_O_2_ was disclosed.

It has been reported that Def-1, an antibacterial bee-derived peptide, is one of the major antibacterial compounds found in every type of honey [4,8,40,41]. Apart from its antibacterial/antibiofilm properties, Def-1 contributes to cutaneous wound closure and promotes wound re-epithelization [42]. Nevertheless, we showed in this study, and elsewhere [10], that destroying of Def-1 in honey by protease digestion did not affect the overall antibacterial activity of blossom and honeydew honeys. Similarly, Stagos et al. (2018) demonstrated that antibacterial activity of proteinase K-treated Greece, mainly polyfloral, honeys did not differ from the untreated ones [13]. These findings suggest that Def-1 and other protein components of bee or botanical origin do not contribute (or their contribution is negligible) to the antibacterial effects of the honey. Generally, these observations undoubtedly have some advantages from a clinical point of view and support the use of medical-grade honey in clinical practice for healing of chronic and infected wounds. Chronic wounds remain in the inflammatory stage of healing, where degradation of the extracellular matrix by proteases, particularly matrix metalloproteinases, occurs [43]. In addition, chronic wounds are commonly colonized by invading bacteria, which also secrete proteases that contribute to the overall high proteolytic activity in the wound environment. Defensins and other antimicrobial peptides are susceptible to degradation and inactivation by wound or bacterial proteases [44]. Our findings suggest that proteolytic degradation and inactivation of biologically active enzymes and peptides in honey, after its application to wounds, will most likely not be connected with decreasing honey antibacterial efficacy.

Currently, the use of honey in wound care is less mainstream, mainly due to: (a) the low quality of clinical evidence supporting the use of honey in wound and burn treatment [45]; (b) the unclear overall impact of honey on wound healing mechanisms [46]; and (c) the unclear mechanisms of action of different types of honey in wound healing. The present study, together with our recent observations [10], showed that antibacterial effects of honeydew and blossom honeys is H_2_O_2_-dependent, and their antibacterial efficacy can be—at least for honeydew, polyfloral honeys, and some monofloral honeys—as effectiveas or more effective than manuka honey. Despite manuka honey having some advantages compared to other honeys, such as the existing knowledge of the mechanism of its action through methylglyoxal (it enables its standardization). It is important to select another source of honey with proven antibacterial activity that will serve as a base for new kinds of medical-grade honey.

In conclusion, 57 tested blossom honey samples showed different antibacterial efficacy against *S. aureus* and *P. aeruginosa* as follows: wildflowers honeys > acacia honeys > rapeseed honeys. Four selected honeys of each honey type were subjected to detailed characterization of the antibacterial activity. This activity was markedly reduced by treatment with catalase but remained stable after complete protein digestion, suggesting that the antibacterial activity of blossom honeys is not directly dependent on GOX-mediated production of H_2_O_2_ or the presence of Def-1. Lastly, and most importantly, we demonstrated that the overall antibacterial activity of honeys was correlated with the level of total H_2_O_2_ present already in ripened honeys and with total polyphenol content.

## 4. Materials and Methods

### 4.1. Honey Samples

Honey samples (*n* = 57) collected in 2016 were received from beekeepers from several regions of Slovakia. Upon receiving, they were immediately stored in plastic containers at 4 °C in the dark. Identification of the floral source of the honey was carried out by a melissopalynological analysis, as reported above. Medical-grade honey, manuka honey MGO 550, was purchased from Manuka Health (Newmarket, Auckland, New Zealand). Artificial honey was prepared as described elsewhere [47] by dissolving 39 g d-fructose, 31 g d-glucose, 8 g maltose, 3 g sucrose, and 19 g distilled water.

### 4.2. Microorganisms

The antibacterial activity of honey samples was assessed against the isolates *Pseudomonas aeruginosa* CCM1960 and *Staphylococcus aureus* CCM4223, obtained from the Department of Medical Microbiology, Slovak Medical University (Bratislava, Slovakia).

### 4.3. Strains, Vectors, Enzymes, and Reagents Used for GOX Gene Cloning

*Escherichia coli* strain DH5α (ThermoFisher Scientific, Loughborough, UK) was used as the host for GOX cDNA cloning. *E. coli* Rosetta-gami (DE3) was purchased from Novagen (Darmstadt, Germany) and used as the host for expression of the heterologous protein. Bacteria were grown in Luria–Bertani (LB) medium or LB agar plates with appropriate antibiotics at 37 °C. *E. coli* cells were transformed using a standard heat-shock method. Plasmid pET28b(+) (Novagen) was chosen for the construction of expression plasmid. Phusion High-Fidelity PCR Master Mix, T4 DNA ligase, and restriction enzymes were purchased from New England BioLabs (Hitchin, UK), while other chemicals were purchased from Sigma-Aldrich (Taufkirchen, Germany) or Promega (Madison, WI, USA).

### 4.4. Construction of the GOX Expression Plasmid and Expression of Recombinant GOX (rGOX)

The cDNA encoding mature honeybee GOX were amplified using following primers: sense primer 5′-GGCGCGCGCCCATGGAGCCGTGCCAGCGC-3′; antisense primer 5′-GGCGCGCGCGGCCGCCAGTAATTGAAGTAATTCTCC-3′. The amplified cDNA fragment was purified, digested with *Nco*I and *Not*I, then ligated into the pET28b(+) vector, which was digested with the same restriction enzymes, to construct the expression plasmid pET28-GOX. The resulting plasmid was transformed into DH5α *E. coli* and cDNA verified by DNA sequencing (GATC Biotech, Constance, Germany). The correct plasmid encoded a translational fusion protein containing the mature protein sequence of bee GOX, followed by a 6x His tag.

To express rGOX, an *E. coli*/pET28-GOX clone was inoculated into 15 mL LB medium supplemented with kanamycin (30 μg/mL) and chloramphenicol (50 μg/mL), then cultured at 37 °C overnight with shaking at 250 rpm. The overnight culture was diluted 100-fold into fresh LB medium (the total volume for expression of rGOX was 1200 mL) with mentioned antibiotics. When the cell culture reached the absorbance at 600 nm about 0.5 (mid-exponential phase), the expression of rGOX was induced by the addition of IPTG to a final concentration of 1.0 mM. After 4 h of incubation at 37 °C and 250 rpm, the harvested cells were immediately centrifuged at 4000 rpm for 10 min.

### 4.5. Purification of rGOX

The rGOX was isolated from crude bacterial lysate under denaturing conditions by affinity chromatography employing Ni–NTA (Qiagen, Hilden, Germany). A soluble protein fraction from the harvested cells was extracted using B-Per Reagent (ThermoFisher Scientific), according to the manufacturer’s instructions. After centrifugation at 14,000 rpm for 20 min, the supernatant was removed and the pellet (insoluble protein fraction) was solubilized in denaturation buffer (6 M urea, 0.02 M Tris-HCl, 0.5 M NaCl, pH 8.0). The obtained solution was loaded onto the Ni–NTA column pre-equilibrated in denaturation buffer. The column was washed in denaturing buffer supplemented with 50 mM imidazole and pure rGOX was eluted with denaturing buffer supplemented with 250 mM imidazole. After Ni-NTA chromatography, purified rGOX was subjected to exchange buffer using a PD-10 desalting column (GE Healthcare, Little Chalfont, UK), eluted with 4 M urea solution and stored at −20 °C. The purity of the prepared rGOX was determined by 12% SDS PAGE, and total protein content was measured using the Quick Start Bradford protein assay (Bio-Rad, Hercules, CA, USA) as described in the instruction manual.

### 4.6. Melissopalynological Analysis

Qualitative and quantitative analyses of the honey sediment were performed according to the method proposed by Von der Ohe et al. [48]. Briefly, ten grams of honey were dissolved in 20 mL of distilled water and then centrifuged for 5 min at 3000× *g*. The pellet, after washing with distilled water, was spread on a microscope slide, dried at 40 °C, and colored by the addition of some drops of glycerin jelly enriched with 0.1% basic fuchsine ethanol solution. The slide was covered with a cover slip and examined under optical microscope. Pollen grains were identified and counted in groups of 100, following five parallel equidistant lines uniformly distributed from one edge of the cover slip to the other. For each pollen type, the relative frequency, or rather the percentage amount compared to the total number of counted pollen grains, was calculated. Moreover, non-identified grains and honeydew indicators were counted. Honey botanical origin was determined on the basis of the relative frequencies of pollen types identified in the microscopic analysis, as reported by Di Marco and co-workers [49].

### 4.7. Determination of Honey Antibacterial Activity

The antibacterial efficacy of the honey samples was evaluated with a MIC assay as described by Bucekova et al. (2014) [20]. Briefly, overnight bacterial culture was suspended in phosphate-buffered saline (PBS), pH 7.2, and the turbidity of the suspension was adjusted to 10^8^ colony-forming units (CFU)/mL and diluted with Mueller–Hinton broth (MHB) medium (pH 7.3 ± 0.1) to a final concentration of 10^6^ CFU/mL. Then, 10 μL aliquots of suspension were inoculated into each well of a sterile 96-well polystyrene U-shape plates (Sarstedt, Nümbrecht, Germany). The final volume in each well was 100 μL, consisting of 90 μL of sterile medium or diluted honey and 10 μL of bacterial suspension. After 18 h of incubation at 37 °C and 1000 rpm, bacterial growth inhibition was determined by monitoring the optical density at 490 nm. The MIC was defined as the lowest concentration of honey that completely inhibits bacterial growth. All tests were performed in triplicate and repeated three times.

Serial dilutions of each honey sample were prepared from a 50% honey solution (*w*/*w* in MHB medium) by dilution with the medium, resulting in final concentrations of 40, 35, 30, 25, 20, 18, 16, 14, 12, 10, 8, 6, and 4%. Similarly, serial dilutions of artificial honey were prepared from a 50% honey solution as described above.

MBC values of honey samples were also evaluated [50]. The viability of bacteria in wells with no turbidity was determined by spreading 100 μL onto an MHB agar plate and incubating at 37 °C for 24 h. The lowest concentration of honey that resulted in no survival of viable bacteria was recorded as MBC.

### 4.8. Determination of GOX Content

Aliquots (15 μL) of 50% (*w*/*w*) honey solution were resolved by SDS-PAGE using a Mini-Protean II electrophoresis cell (Bio-Rad). The proteins were transferred onto a 0.22-μm nitrocellulose Advantec membrane (Sigma-Aldrich) in 12 mM Tris, 95 mM glycine and 20% methanol using the wet blotting procedure. The membrane was blocked for 1 h in a Tris-buffered saline-Tween (TBST) buffer (50 mM Tris-HCl, pH 7.5, 200 mM NaCl, and 0.05% Tween 20) containing 5% non-fat dried milk and incubated overnight with a rabbit polyclonal antibody against honeybee GOX (1:2000 in TBST) which was prepared by GenCust Europe (Dudelange, Luxembourg). After washing with TBST, the membranes were incubated for 2 h in blocking buffer containing goat anti-rabbit horseradish peroxidase-linked antibodies (1:2500 in TBST; Promega). Immunoreactive bands were detected in solution containing dissolved SigmaFast 3,3-diaminobenzidine tablets (Sigma-Aldrich), and specific bands were quantified by densitometry using ImageJ software (NIH, version 1.52a, Bethesda, MD, USA).

### 4.9. Determination of H_2_O_2_ Content

H_2_O_2_ content in honey samples was determined using Megazyme GOX assay kit (Megazyme International Ireland Ltd, Bray, Ireland), which is based on H_2_O_2_ release. As a standard, H_2_O_2_ diluted to 9.8–312.5 μM was used. 40% (*w*/*w*) honey solutions in 0.1 M potassium phosphate buffer (pH 7.0) were prepared and immediately measured. Each honey sample and H_2_O_2_ standard was tested in duplicate in a 96-well microplate. The absorbance of reaction was then measured at 510 nm using a Synergy HT microplate reader (BioTek Instruments, Winooski, VT, USA).

### 4.10. Determination of Total Polyphenolic Content

Total polyphenolic content was determined with a Folin–Ciocalteu Phenolic Content Quantification Assay Kit (BioQuoChem, Llanera, Spain) in a 20% (*w*/*v*) honey solution in a 96-well microplate according to the manufacturer’s instructions. Gallic (GAE) acid was used as the reference standard and results were expressed as GAE equivalents (μg GAE/g of honey). Absorbance was measured at 700 nm at 37 °C.

### 4.11. Enzymatic Treatment of Honey Samples with Catalase and Proteinase K

Diluted honey samples (50% *w*/*w* in MHB medium) were treated with catalase (2000–5000 U/mg protein; Sigma-Aldrich) at a final concentration of 1000–2500 U/mL at room temperature for 2 h, or proteinase K (30 U/mg; Promega) at a final concentration of 50 μg/mL at 37 °C for 30 min. Aliquots (15 μL) of proteinase-K treated samples were separated by SDS-PAGE using a Mini-Protean II electrophoresis cell (Bio-Rad) and treatment efficacy assessed after gel staining with Coomassie Brilliant Blue R-250. Catalase- and proteinase K-treated honey samples were then immediately used in the antibacterial assay to determine MIC values against *S. aureus* and *P. aeruginosa*.

### 4.12. Statistical Analysis

One-way ANOVA and Tukey’s multiple comparison tests were used to evaluate antibacterial efficacy of different botanical types of honey. The Pearson correlation test was used for correlation analysis between antibacterial activity/GOX content and H_2_O_2_ content or total polyphenol content in honeys. The data are expressed as mean values with the standard deviation (SD). Data with *P* values smaller than 0.05 were considered as statistically significant. All statistical analyses were performed using GraphPad Prism (GraphPad Software Inc., version 5, La Jolla, CA, USA).

## Figures and Tables

**Figure 1 molecules-24-01573-f001:**
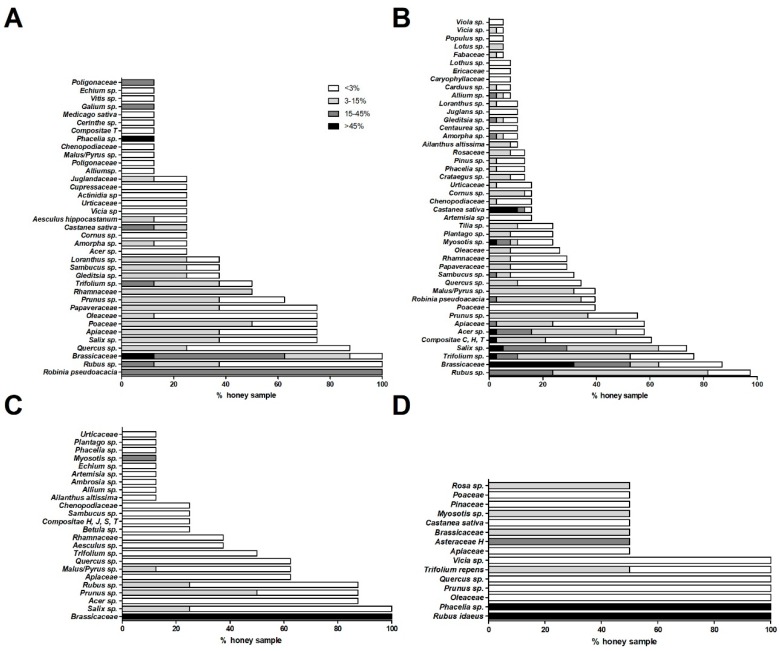
Pollen spectra of different types of honey-acacia (**A**), wildflower (**B**), rapeseed (**C**), and phacelia and raspberry (**D**) are given. For each graph, the percentage of honey samples showing different pollen types (family, genus, or species) with relative abundance frequency (>45%; 15–45%; 3–15%; and <3%) are shown.

**Figure 2 molecules-24-01573-f002:**
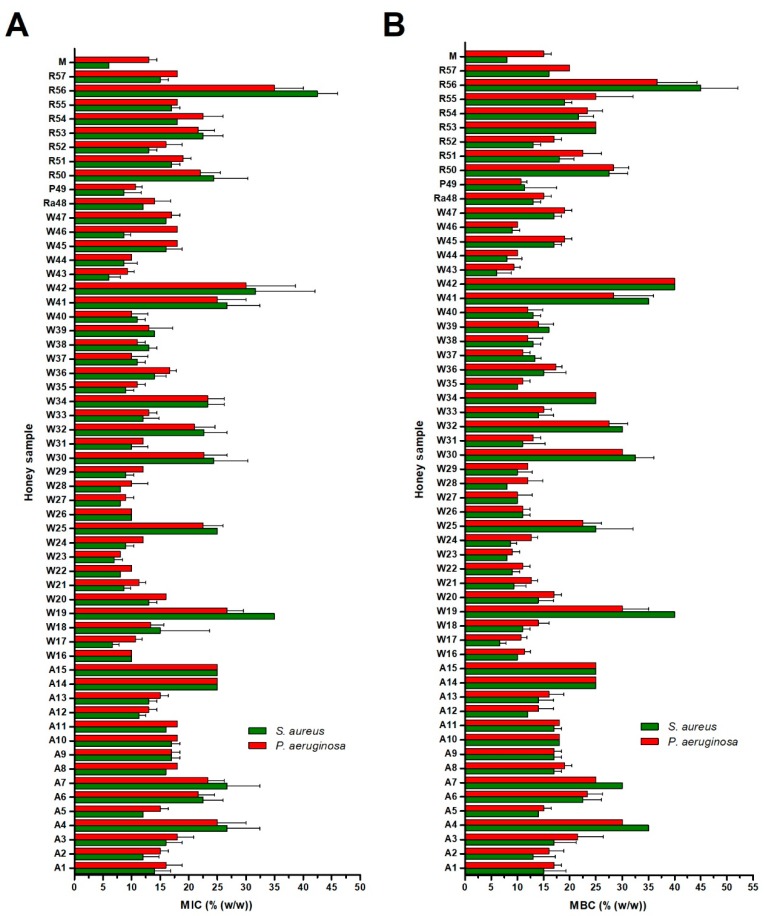
Antibacterial activity of blossom honey samples (*n* = 57) and medical-grade manuka against *Staphylococcus aureus* and *Pseudomonas aeruginosa* isolates. Activity was determined with a minimum inhibitory concentration (MIC) (**A**) and minimum bactericidal concentration (MBC) (**B**) assay. The MIC and MBC were defined as the lowest concentration of honey solution (%) inhibiting bacterial growth and killing the bacteria, respectively. Honeys: M—manuka honey, R—rapeseed, P—phacelia, Ra—raspberry, W—wildflower, A—acacia. The data are expressed as the mean values with standard deviation (SD).

**Figure 3 molecules-24-01573-f003:**
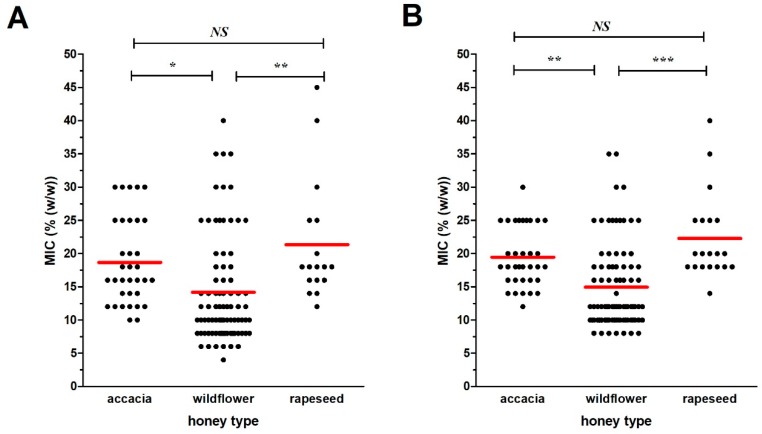
Comparison of overall antibacterial efficacy of different honey types. Antibacterial activity of acacia (*n* = 15), wildflower (*n* = 32), and rapeseed (*n* = 8) honeys was determined against (**A**) *Staphylococcus aureus* and (**B**) *Pseudomonas aeruginosa*. Activity was determined with a minimum inhibitory concentration (MIC). Red line represents the mean of all measured data. Differences among groups of samples belonging to different honey types were analyzed by ANOVA. Asterisks indicate a significant difference between the examined honey types, * *P* < 0.05, ** *P* < 0.01, *** *P* < 0.001, NS-non-significant.

**Figure 4 molecules-24-01573-f004:**
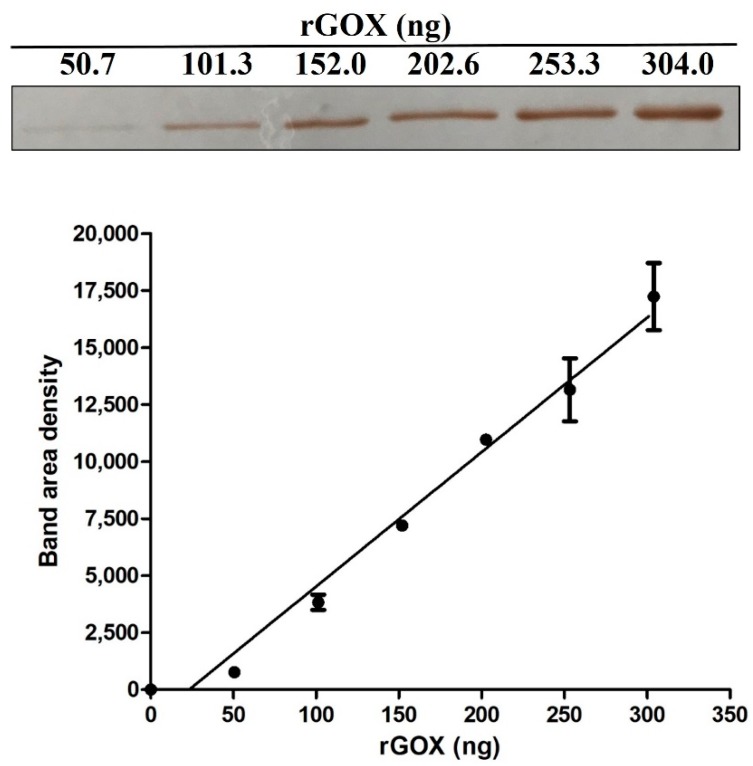
Quantitative profile of recombinant glucose oxidase (rGOX) on immunoblot used for GOX quantification in honeys. rGOX was prepared by a heterologous expression in *Escherichia coli*. Specific immunoreactive bands on membrane were quantified by densitometry using ImageJ software (NIH, USA). The graph shows good correlation between increasing amounts of rGOX and densitometry signals on the created calibration curve.

**Figure 5 molecules-24-01573-f005:**
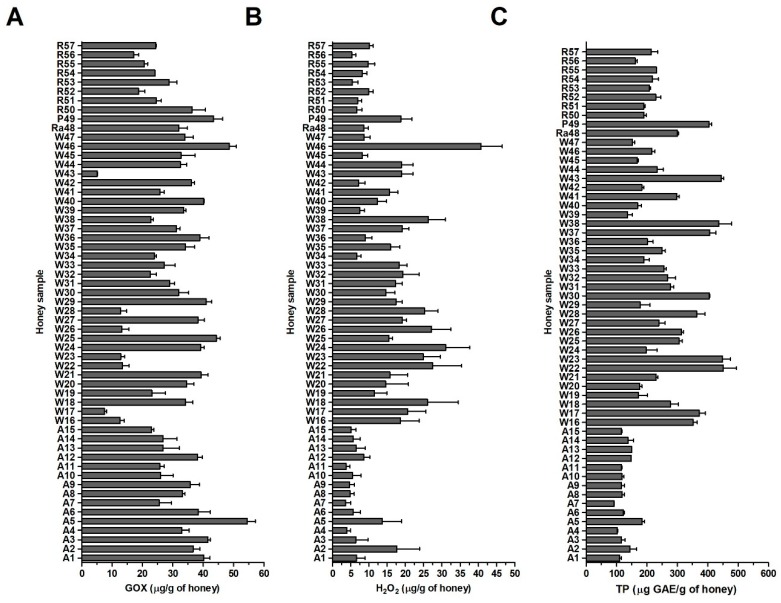
Glucose oxidase (GOX), hydrogen peroxide (H_2_O_2_) and total polyphenol contents (TP) in blossom honey samples (*n* = 57). (**A**) GOX content was determined using a semi-quantitative immunoblot assay (**B**) H_2_O_2_ content was measured in diluted honey samples (40% (*w*/*w*), without previous incubation) with a modified GOX assay kit (**C**) TP content was determined with a Folin Ciocalteu Phenolic Content Quantification Assay Kit in 20% (*w*/*v*) honey solutions. Gallic acid (GAE) was used as the TP reference standard compound and results are expressed as GAE equivalents (μg GAE/g of honey). The data represent the mean values with standard deviation (SD).

**Figure 6 molecules-24-01573-f006:**
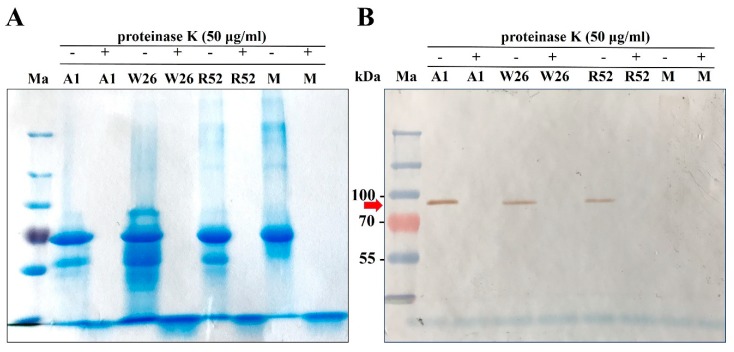
Protein profile of untreated and proteinase-K-treated selected blossom honey samples and manuka honey (M) and immunoblot analysis of glucose oxidase (GOX). Aliquots (15 μL) of 50% (w/w) honey solution in MHB medium with or without proteinase K treatment were separated by SDS-PAGE. Gel was stained with Coomassie Brilliant Blue R-250 (**A**) or immunoblotted with a polyclonal antibody against GOX (**B**). Ma, protein prestained marker. Red arrow indicates GOX with a molecular weight of 84 kDa.

**Figure 7 molecules-24-01573-f007:**
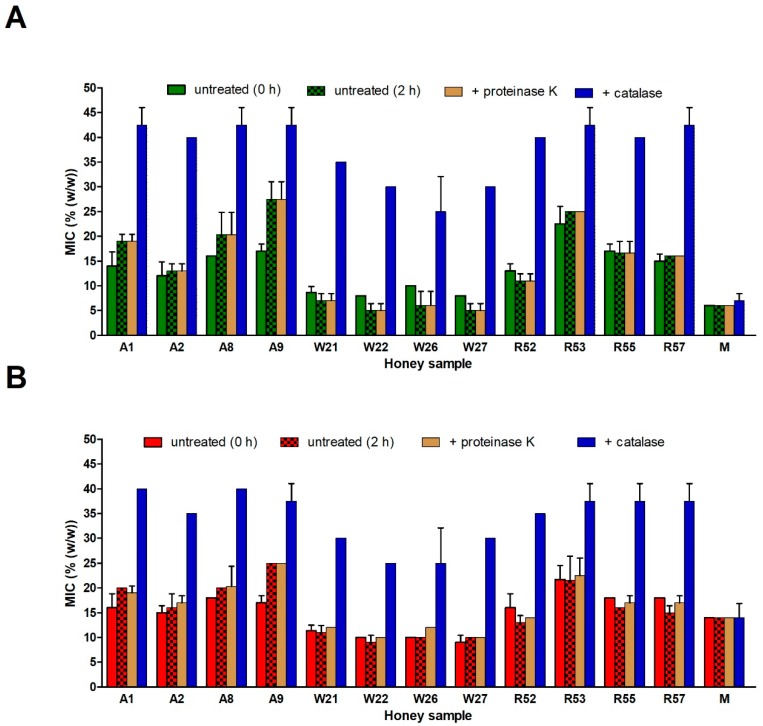
Antibacterial activity of selected blossom honeys (*n* = 12) and medical-grade manuka honey following catalase and proteinase K treatment against (**A**) *Staphylococcus aureus* and (**B**) *Pseudomonas aeruginosa* isolates. The 50% (*w*/*v*) honey solutions were treated with catalase at room temperature (RT) for 2 h or with proteinase K at 37 °C for 30 min. As controls, antibacterial activity of untreated 50% (*w*/*w*) honey solutions are given, both not incubated (0 h) and incubated at RT for 2 h. The antibacterial activity was determined with a MIC assay. The MIC was defined as the lowest concentration of honey solution (%) inhibiting bacterial growth. M, manuka honey. The data are expressed as the mean values with standard deviation (SD).

**Figure 8 molecules-24-01573-f008:**
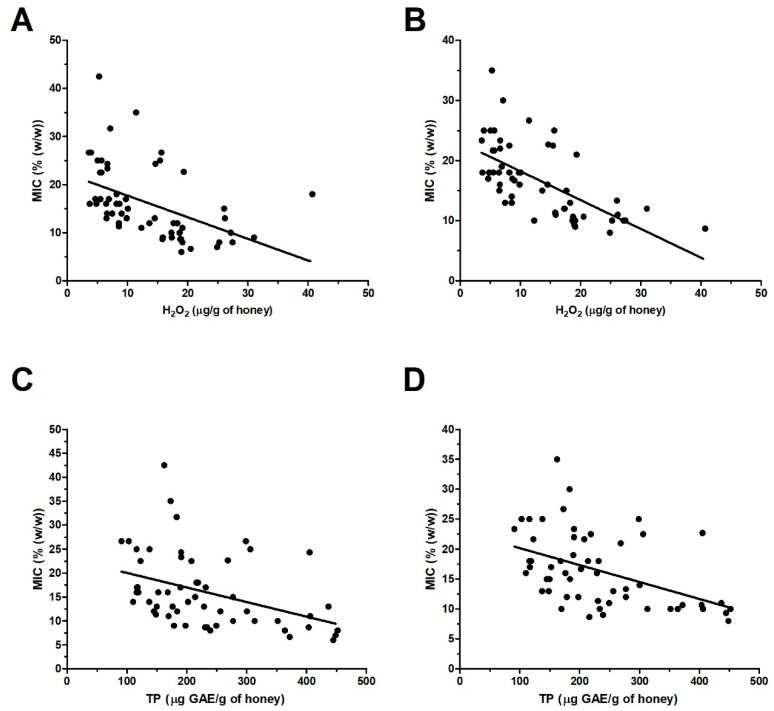
Graphical representation of the relationship between the content of H_2_O_2_ (**A**,**B**) or total polyphenol (TP) content (**C**,**D**) and the overall antibacterial activity of blossom honey samples (*n* = 57) against *Staphylococcus aureus* (**A**,**C**) and *Pseudomonas aeruginosa* (**B**,**D**) isolates. A Pearson correlation test was used for correlation analysis.

**Figure 9 molecules-24-01573-f009:**
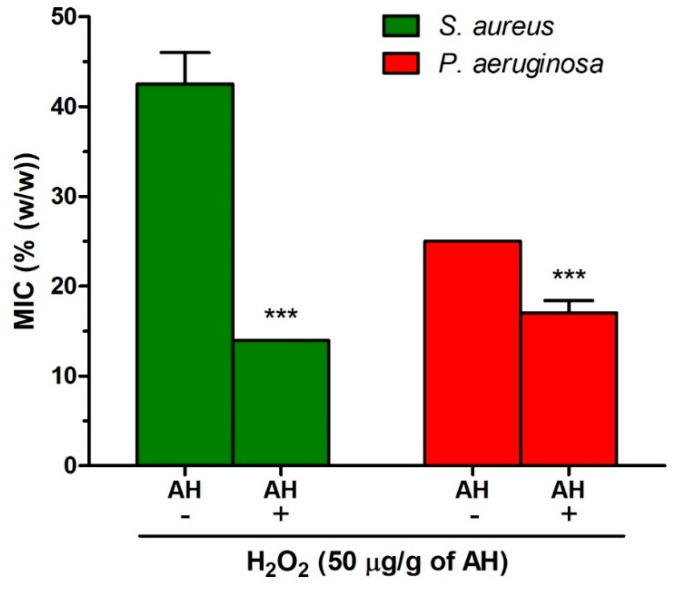
Antibacterial activity of artificial honey (AH) and AH enriched in hydrogen peroxide H_2_O_2_ at concentration of 50 μg/g of AH against *Staphylococcus aureus* and *Pseudomonas aeruginosa* isolates. The antibacterial activity was determined with a MIC assay. The MIC was defined as the lowest concentration of honey solution (%) inhibiting bacterial growth. The data are expressed as the mean values with standard deviation. Differences between honey groups were analyzed by *t*-test. Asterisks indicate a significant difference between control AH and AH enriched in H_2_O_2_, *** *P* < 0.001.

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
