# Peer review of "Antibacterial Activity of Different Blossom Honeys: New Findings"

_molecules, 2019, doi:10.3390/molecules24081573_

Round 1
Reviewer 1 Report
The role of glucose oxidase in antibacterial activity of blossom honey: breaking the dogma
Summary
The manuscript offers a concept that hydrogen peroxide in honey is not a product of glucose oxidase reaction but rather is formed via other, non-enzymatic process. The conclusion is based on the observed lack of correlations between, glucose oxidase activity, glucose oxidase amount in different honey samples and concentration of hydrogen peroxide in these samples. While this work supported previous literature data that H2O2 is the main contributor to the antibacterial activity of honey, the authors contest the concept that glucose oxidase is responsible for hydrogen peroxide generation. A complete protein digestion of honey with proteinase K did not changed its antibacterial activity. The authors’ conclusion drawn from these experiments suggests that “the antibacterial activity of blossom honeys is not –dependent on GOX-mediated production of H2O2 or the presence of Def-1.”
The conclusions of this quantitative research are based on imprecise and incomplete data.
1. Different assays conditions: GOX activity was investigated at 20% honey dilution, measurements of hydrogen peroxide concentration was conducted at 40% honey dilutions, the concentration of GOX protein was estimated from semi-quantitative Western blotting. The values generated from these different methodologies conducted under different conditions were used for statistical correlations.
2. There is lack of data/ figure on the changes GOX activity upon honey dilutions. Why GOX activity was measured at 20% honey dilution? One of limiting factors of GOX activity is sugar concentration (honey dilution) and pH.
3. There is lack of data/figure on the changes in hydrogen peroxide concentration upon honey dilutions. The H2O2 production depends on honey sugar concentration and increases proportionally to the honey dilutions only to a certain dilution point. After reaching the maximum, H2O2 concentration rapidly declines with further dilution. The maximum of H2O2 production is different for different honeys. Since MIC correlates with the peak of H2O2 production, therefore the measurements of H2O2 production should be then made at the MIC dilution, which was shown in this manuscript to range between 6.0 to 45.0% against S. aureus and between 8.0 to 40.0% against P. aeruginosa (and not arbitrary at 40%).
4. Data from quantitative Western blot should be presented. Is there any evidence that the GOX protein concentration is proportional to GOX activity at 20% honey dilution?
5. RE: The role of honey proteins in the antibacterial effect of honeys. There is lack of convincing evidence that protein digestion by proteinase K in honey was complete. The protein concentration has not been measured before and after digestion. No other documentation of protein removal from honey was provided.
6. The Results section of the manuscript includes the paragraph describing in details pollen profile in four varieties of honey. It occupies significant part of the manuscript (1/3 of the Results) but brings little to support the main concept. On the other hand, there has been not enough experimental details listed above that would convincingly validate the statement that the antibacterial activity of blossom honeys is not –dependent on GOX-mediated production of H2O2.
Author Response
Thank you for your valuable comments and suggestions.
1. Different assays conditions: GOX activity was investigated at 20% honey dilution, measurements of hydrogen peroxide concentration was conducted at 40% honey dilutions, the concentration of GOX protein was estimated from semi-quantitative Western blotting. The values generated from these different methodologies conducted under different conditions were used for statistical correlations.
Thank you for your comment. An assay for determination of GOX activity as well as assay for determination of total hydrogen peroxide was carefully optimised with various honey samples (dark, light, untreated, heated etc.). In both assays, the samples were diluted in buffered solution with pH 7.0 to avoid of different pH value of each honey. Furthermore, medium for cultivation bacteria (MHB medium) with the similar pH value (7.2), where honey was diluted, was used.
In case of estimation of GOX content we chose a semi-quantitative method using Western blotting which was established for this work but it can be used for every type of honey. All used methods were carefully optimised and it is not possible to perform quantification of GOX and determination of total hydrogen peroxide under the same conditions due to different approach and final concentration of desired molecules.
2. There is lack of data/ figure on the changes GOX activity upon honey dilutions. Why GOX activity was measured at 20% honey dilution? One of limiting factors of GOX activity is sugar concentration (honey dilution) and pH.
We optimised GOX assay when honey was used as a sample. As we mentioned above, the pH value of all samples was adjusted to 7.0 and GOX activity in honey solution was determined exactly after 20 min of incubation. We tested various honey dilutions and the optimal concentration for this assay was 20% honey solution, which was the best suited in a range of calibration curve. However, we admit that estimation of GOX activity in honey with different molecules and interactions is complicated and might be misleading. The principle of the method is based on measurement of accumulated hydrogen peroxide in honey solution in 20 min of incubation. Due to potential additional source of hydrogen peroxide generation, the results did not actually reflect the sole enzymatic activity of GOX. Therefore, we decided to remove the whole data set of GOX activity from manuscript.
3. There is lack of data/figure on the changes in hydrogen peroxide concentration upon honey dilutions. The H2O2 production depends on honey sugar concentration and increases proportionally to the honey dilutions only to a certain dilution point. After reaching the maximum, H2O2 concentration rapidly declines with further dilution. The maximum of H2O2 production is different for different honeys. Since MIC correlates with the peak of H2O2 production, therefore the measurements of H2O2 production should be then made at the MIC dilution, which was shown in this manuscript to range between 6.0 to 45.0% against S. aureus and between 8.0 to 40.0% against P. aeruginosa (and not arbitrary at 40%).
We totally agree that different honeys produce different level of hydrogen peroxide. The maximum levels of accumulated H2O2 that occurred in honey solutions were found in solutions diluted to concentrations between 30 and 50 % (Bang et al. 2003). We focused on overall capability of each honey sample to produce hydrogen peroxide at honey concentration of 40%.
We assume that the profile (not values) of total hydrogen peroxide production among the samples will be similar at their MIC values (supported by results of study by Brudzynski et al, 2011). Furthermore, statistical analysis clearly showed a strong significant correlation between the concentration of H2O2 and antibacterial activity. Your suggested procedure is rationale but does not account also the contribution of bacterial H2O2 destroying system and interaction of hydrogen peroxide with polyphenols that can affect the MIC. In further study we will take into account your suggestion.
4. Data from quantitative Western blot should be presented. Is there any evidence that the GOX protein concentration is proportional to GOX activity at 20% honey dilution?
As we mentioned above, we decide to remove the whole data set of GOX activity from manuscript. We added a calibration curve (band area density vs. amount of GOX) and one demonstrative western blot where rGOX at different amount was visualised. Please, see revised manuscript.
5. RE: The role of honey proteins in the antibacterial effect of honeys. There is lack of convincing evidence that protein digestion by proteinase K in honey was complete. The protein concentration has not been measured before and after digestion. No other documentation of protein removal from honey was provided.
The digestion of all protein content in honey samples treated with proteinase K was checked by SDS-PAGE. Proteinase K is a very robust and effective protease which works in different pH and temperature. Same procedure was used in case of honeydew honey samples where this process is documented by figures (Bucekova et al, 2018 in Scientific Reports). To provide more evidence, we added a new Figure into the manuscript where one sample of each botanical group of blossom honeys (1 acacia, 1 wildflower and 1 rapeseed honey) and manuka honey were subjected on SDS PAGE before and after proteinase K digested.
6. The Results section of the manuscript includes the paragraph describing in details pollen profile in four varieties of honey. It occupies significant part of the manuscript (1/3 of the Results) but brings little to support the main concept. On the other hand, there has been not enough experimental details listed above that would convincingly validate the statement that the antibacterial activity of blossom honeys is not –dependent on GOX-mediated production of H2O2.
We significantly shortened the results from pollen analysis. It is very important to know what type of honey is being tested. We agree that we did not provide a clear evidence that hydrogen peroxide generated by enzymatic reaction. On the other hand, we provide some evidences that the content of GOX did not correlate with antibacterial activity and hydrogen peroxide can be generated by an alternative way. Furthermore, polyphenol content might play an important role in antibacterial action. Some of our “strong” expressions were reformulated.
Reviewer 2 Report
The manuscript by Bucekova et al investigated the role of glucose oxidase (GOX) in antibacterial activity of blossom honey. The topic is interesting and the manuscript is easy to read; however, there are issues with the methods, resulting several conclusions that were made without sufficient data in this study.
Their key conclusion is "the antibacterial activity of blossom honeys is not dependent on GOX content", and the experiment to support this conclusion is to treat honey samples with proteinase K to break down GOX and then test for antibacterial activity. This kind of experiment cannot help to make this conclusion. Providing that proteinase K can destroy all GOX variants, the ultimate removal of GOX does not affect the existing H2O2 that was produced by GOX long time before the proteinase K treatment. Therefore, H2O2 was still there, and so the antibacterial activity was still observed in the absence of GOX or independent of GOX. The more proper study would be: (1) hydrolyzing existing proteins by proteinase K, then (2) removing existing H2O2 by catalase, and then (3) adding rGOX to honeys see if it could boost antibacterial activity.
Other concerns about their methods:
- The author concluded "As expected, honey samples were more effective against the Gram-positive S. aureus compared with the Gram-negative P. aeruginosa" (lines 157-158), unfortunatly statistic analysis was not conducted for this comparison. JPCA (principle component analysis) or a similar statistic analysis should be done for data shown in Figure 2.
- Line 195: "rGOX in denaturing solution (4 M urea, 20 mM Tris pH 8.0) with 70%purity" Generally, the purity for a recombinant protein should be higher than 90 %. The authors should provide the reason or at least speculate for this low purity
- Line 194: "No cross-reaction with other proteins could be detected". There are millions of proteins, how did the authors know the antibody don't bind those proteins?
- Line 211: 'GOX activity was determined in 20% (w/v) honey solutions with a GOX assay kit", how did the authors ensure other components in honey solutions, besides GOX, did not interfere with the GOX assay?
Author Response
Thank you for your valuable comments and suggestions.
Their key conclusion is "the antibacterial activity of blossom honeys is not dependent on GOX content", and the experiment to support this conclusion is to treat honey samples with proteinase K to break down GOX and then test for antibacterial activity. This kind of experiment cannot help to make this conclusion. Providing that proteinase K can destroy all GOX variants, the ultimate removal of GOX does not affect the existing H2O2 that was produced by GOX long time before the proteinase K treatment. Therefore, H2O2 was still there, and so the antibacterial activity was still observed in the absence of GOX or independent of GOX. The more proper study would be: (1) hydrolyzing existing proteins by proteinase K, then (2) removing existing H2O2 by catalase, and then (3) adding rGOX to honeys see if it could boost antibacterial activity.
Thank you for your valuable comments. We agree that during the nectar processing into a honey, GOX is able to produce certain level of hydrogen peroxide, that can be found in 100% honey. On the other hand, the level of accumulated hydrogen peroxide in 100% is very low and it is not responsible for overall antibacterial activity of honey. Proteinase K is a very robust and effective protease working in a broad spectrum of pH and temperature. We were able to observe a complete honey protein digestion within a period of 30 min. Our preliminary unpublished results clearly show that production of hydrogen peroxide (time-production curve) in proteinase K-treated and untreated honey samples is statistically non-significant, and there has to be other source of hydrogen peroxide. We agree with your suggestion and we think in the same way. Currently, we purify a native GOX enzyme and also try to prepare soluble and active recombinant enzyme in order to add GOX enzyme into the artificial honey and also natural raw honeys and provide clear evidence about the role of GOX in antibacterial activity of honey. This will be topic of our further study.
- The author concluded "As expected, honey samples were more effective against the Gram-positive S. aureus compared with the Gram-negative P. aeruginosa" (lines 157-158), unfortunatly statistic analysis was not conducted for this comparison. JPCA (principle component analysis) or a similar statistic analysis should be done for data shown in Figure 2.
Based on the present results in Figure 2, we concluded that S. aureus was more sensitive to honey than P. aeruginosa. We did not aim to perform statistical analysis just between two bacterial strains. In fact, some of the antibacterial components such as defensin-1 peptide found in honey is solely effective against Gram-positive bacteria. It is not necessary in this manuscript to statistically analyse the difference between used two bacterial strains.
- Line 195: "rGOX in denaturing solution (4 M urea, 20 mM Tris pH 8.0) with 70%purity" Generally, the purity for a recombinant protein should be higher than 90 %. The authors should provide the reason or at least speculate for this low purity.
Generally, recombinant proteins produced by synthesis (synthetic proteins) are at high purity (over 90%). Recombinant proteins prepared in the live expression system (prokaryotic or eukaryotic) need to be purified through several steps. Purification of GOX was done under denaturing conditions (6 M urea) since GOX was expressed as we expected as an insoluble molecule in Escherichia coli. There is limited options for further purification of denatured recombinant protein. Furthermore, the level of expression was not high. Due to these facts, we worked with rGOX at purity of 70%.
Furthermore, our polyclonal antibody is able to recognise the GOX only when is denatured.
- Line 194: "No cross-reaction with other proteins could be detected". There are millions of proteins, how did the authors know the antibody don't bind those proteins?
In fact, proteins/polypeptides in honey are at very low concentrations and the most abundant are major royal jelly proteins (MRJP1-5). No cross-reaction with other honey proteins was detected. We corrected accordingly.
- Line 211: 'GOX activity was determined in 20% (w/v) honey solutions with a GOX assay kit", how did the authors ensure other components in honey solutions, besides GOX, did not interfere with the GOX assay?
We optimised GOX assay when honey was used as a sample. As we mentioned above, the pH value of all samples was adjusted to 7.0 and GOX activity in honey solution was determined exactly after 20 min of incubation. We tested various honey dilutions and the optimal concentration for this assay was 20% honey solution, which was the best suited in a range of calibration curve. However, we admit that estimation of GOX activity in honey with different molecules and interactions is complicated and might be misleading. The principle of the method is based on measurement of accumulated hydrogen peroxide in honey solution in 20 min of incubation. Due to potential additional source of hydrogen peroxide generation, the results did not actually reflect the sole enzymatic activity of GOX. Therefore, we decided to remove the whole data set of GOX activity from manuscript.
Reviewer 3 Report
The antibacterial activity of honey is a complex phenomenon resulting of glucose oxidase activity, some specific phytochemicals, high sugar concentration, acidic pH, water activity, osmotic properties. However it is intensively studied, and since now it is still unclear. The study is focused on the role of glucose oxidase (GOX) in formation of antibacterial activity of 57 samples of nectar honeys from Slovakia. Due to, the antibacterial activity of honey strongly depends on its botanical origin the choise of wildfloral (more commonly named polyfloral or multifloral) honeys to study complicate the results interpretation as among all honey types this kind is the most variable. However, your investigations led to collection of a large pool of valuable data that included establishing of MICs and MBCs of tested honeys against two bacterial spp., as well as GOX-related factors influencing the antibacterial power of honey. However, there is a lack of combination of the obtained results with the assessment of the polyphenols content in these honeys, which would increase the value of the obtained evidence.
This is a solid, well-performed study conducted on a significantly large number of honey samples. However, the structure of the manuscript requires a major reorganization of the material to improve logic and flow of the main ideas. Moreover, the addition of the Principal Component Analysis conducted on all tested variables is needed as a closing paragraph to help with conclusion on the determinants responsible for honey antibacterial activity.
Main comments:
Main comments:
1. The title: I suggest replace "breaking the dogma" by "new evidences" due to your paper did not explain all aspects.
2. The abstract of the manuscript is not well documented to represent the whole study
3. The aim of the research is not well documented, you did not detect the source of hydrogen peroxide source, needs improving
4. Results about melisopalinology are not related with the paper topic, these part of article (lines 76 to 150) needs be reduced. It was also verification of honey variety, not used in other part of study. However the more interesting will be the characterization about honey samples, due to the pollen composition is related with chemical composition of honey (mainly polyphenols are nectar-specific)
5. The content of GOX protein was not correlated with its activity (line 223-225), how do you explain it? If it possible the used assay procedure miss the optimal pH of these enzyme?
6. How did you select the samples to the next part of study (lines 242-244)? Based on its antibacterial activity?
7. How did you control digestion of GOX as defensing-1 protein? In each case you have almost the same results as for raw honey. Why did you use the honey solution in MHB medium (possible interferences)? Is it possible that at these environment the enzymatic protein digestion was not effective?
8. The section 4.3-4.6 described the synthesis of the standard for immunoblotting assay. Was it done specially for this study or not? It is not directly connected with paper topic and disturb the logical form of article.
Minor comments:
Line 61: add the year of publication
Line 72-73: the source?
Line 175: Fig. 2 I suggest you to mark the honey samples as A1, A2…(acacia), it will be more clear presentation (consequently in other figs.)
Line 182: What does mean the dots? They correspond to samples?
Line 221: compare to line 160 where you use 48- raspberry and 49- phacelia
Line 243: add which samples you used as acacia (no. 1,2,8,9)…and describe the selection criteria
Line 251: to catalyze?
Line 295-298: Compare your results to other authors (Sowa et al., 2017 https://doi.org/10.1111/lam.12749; Elbana et al. https://doi.org/10.1016/S2222-1808(14)60504-1)
Line 304: The content of total polyphenols in honeys is variety dependent and in dark honey reaches higher levels (see Dżugan et al, https://doi.org/10.3390/molecules23082069, Alzahrani et al. doi:10.3390/molecules170910540)
Author Response
Thank you for your valuable comments and suggestions
The antibacterial activity of honey is a complex phenomenon resulting of glucose oxidase activity, some specific phytochemicals, high sugar concentration, acidic pH, water activity, osmotic properties. However it is intensively studied, and since now it is still unclear. The study is focused on the role of glucose oxidase (GOX) in formation of antibacterial activity of 57 samples of nectar honeys from Slovakia. Due to, the antibacterial activity of honey strongly depends on its botanical origin the choise of wildfloral (more commonly named polyfloral or multifloral) honeys to study complicate the results interpretation as among all honey types this kind is the most variable. However, your investigations led to collection of a large pool of valuable data that included establishing of MICs and MBCs of tested honeys against two bacterial spp., as well as GOX-related factors influencing the antibacterial power of honey. However, there is a lack of combination of the obtained results with the assessment of the polyphenols content in these honeys, which would increase the value of the obtained evidence.
Based on your comment, we added a new data set from measurement the total polyphenols in all 57 blossom honey samples and investigated the relationship between the polyphenols content and overall antibacterial activity and the concentration of hydrogen peroxide. See revised manuscript.
This is a solid, well-performed study conducted on a significantly large number of honey samples. However, the structure of the manuscript requires a major reorganization of the material to improve logic and flow of the main ideas. Moreover, the addition of the Principal Component Analysis conducted on all tested variables is needed as a closing paragraph to help with conclusion on the determinants responsible for honey antibacterial activity.
The central idea of principal component analysis (PCA) is to reduce the dimensionality of a data set consisting of a large number of interrelated variables while retaining as much as possible of the variation present in the data set. We have been discussing this PCA analysis with statistician and it was not recommended for our study. Correlations obtained by Pearson’s correlation between the investigated parameters were sufficient for this study.
Main comments:
1. The title: I suggest replace "breaking the dogma" by "new evidences" due to your paper did not explain all aspects.
We agree with your comment and we changed the title of ms.
2. The abstract of the manuscript is not well documented to represent the whole study
We re-wrote the abstract. We believe that it summaries all important results from our study.
3. The aim of the research is not well documented, you did not detect the source of hydrogen peroxide source, needs improving.
We reformulated and corrected the sentence.
4. Results about melisopalinology are not related with the paper topic, these part of article (lines 76 to 150) needs be reduced. It was also verification of honey variety, not used in other part of study. However the more interesting will be the characterization about honey samples, due to the pollen composition is related with chemical composition of honey (mainly polyphenols are nectar-specific)
We significantly shortened the results from pollen analysis. It is very important to know what type of honey is being tested. We also added the data set from polyphenol content of all honey samples. As expected wildflower honeys are rich on polyphenols in compare to acacia or rapeseed honeys.
5. The content of GOX protein was not correlated with its activity (line 223-225), how do you explain it? If it possible the used assay procedure miss the optimal pH of these enzyme?
We optimised GOX assay when honey was used as a sample. As we mentioned above, the pH value of all samples was adjusted to 7.0 and GOX activity in honey solution was determined exactly after 20 min of incubation. We tested various honey dilutions and the optimal concentration for this assay was 20% honey solution, which was the best suited in a range of calibration curve. However, we admit that estimation of GOX activity in honey with different molecules and interactions is complicated and might be misleading. The principle of the method is based on measurement of accumulated hydrogen peroxide in honey solution in 20 min of incubation. Due to potential additional source of hydrogen peroxide generation, the results did not actually reflect the sole enzymatic activity of GOX. Therefore, we decided to remove the whole data set of GOX activity from manuscript.
6. How did you select the samples to the next part of study (lines 242-244)? Based on its antibacterial activity?
The selection criterion was antibacterial activity. We mentioned this fact in revised manuscript
7. How did you control digestion of GOX as defensing-1 protein? In each case you have almost the same results as for raw honey. Why did you use the honey solution in MHB medium (possible interferences)? Is it possible that at these environment the enzymatic protein digestion was not effective?
Using a proteinase K enzyme, which is a very robust and effective protease working at different pH value and temperature, we were able to digest all protein content in honey. We digested honey protein in MHB medium since this solution was further used for MIC assay. To provide more evidence, we added a new Figure into the manuscript where one sample of each botanical group of blossom honeys (1 acacia, 1 wildflower and 1 rapeseed honey) and manuka honey were subjected on SDS PAGE before and after proteinase K digested in MHB medium.
8. The section 4.3-4.6 described the synthesis of the standard for immunoblotting assay. Was it done specially for this study or not? It is not directly connected with paper topic and disturb the logical form of article.
It is very important part of the ms. Preparation of GOX standard using heterologous expression system allowed us to construct a calibration curve and established a new semi-quantitative method for determination of total GOX content in honey.
Minor comments:
Line 61: add the year of publication
It was done.
Line 72-73: the source?
It was corrected.
Line 175: Fig. 2 I suggest you to mark the honey samples as A1, A2…(acacia), it will be more clear presentation (consequently in other figs.)
We agree with your suggestion and we corrected all marking of honey samples in all figures
Line 182: What does mean the dots? They correspond to samples?
Dots in Figure 3 are values of individual samples. They can be overlapped when the samples have the same values.
Line 221: compare to line 160 where you use 48- raspberry and 49- phacelia
Data from enzymatic activity (including line 160) were removed from manuscript.
Line 243: add which samples you used as acacia (no. 1,2,8,9)…and describe the selection criteria
We corrected and added the text as you suggested.
Line 251: to catalyze?
It was corrected.
Line 295-298: Compare your results to other authors (Sowa et al., 2017 https://doi.org/10.1111/lam.12749; Elbana et al. https://doi.org/10.1016/S2222-1808(14)60504-1)
Thank you for your suggestion. A selected study (Sowa et al, 2017) was added to revised manuscript.
Line 304: The content of total polyphenols in honeys is variety dependent and in dark honey reaches higher levels (see Dżugan et al, https://doi.org/10.3390/molecules23082069, Alzahrani et al. doi:10.3390/molecules170910540)
Thank you for your suggestions. Both mentioned studies were added to a references list in revised manuscript.
Round 2
Reviewer 1 Report
Several concerns have not been adequately addressed in the revised manuscript.
1. The statement that ”there is no relationship between the level of GOX and the level of H2O2 produced in similarly diluted blossom honeys” (Line 210) remains unsubstantiated. The authors have arbitrarily chosen 20% dilution, although there is lack of data in this manuscript how GOX activity is changing with honey dilutions. Moreover, accumulation of H2O2 is shown to be dependent on the amount of water and is changing with dilution in a non-linear manner approximating an inverted U-shape curve. In addition to that, this curve is different for each honey. The authors have not investigated this relationship and have again chosen arbitrarily 40% dilution without systematic study of H2O2 level in successive dilutions for different honeys. In such situation, changes of GOX level and changes of H2O2 concentration do not represent related changes and cannot be correlated at single selected points. The points of comparison are not corresponding to each other, so correlation cannot be found. The authors should include a systematic investigation of effects of honey dilutions on GOX activity on one hand, and hydrogen peroxide formation with dilutions on the other.
Viscosity, sugar concentration and water content are the main factors influencing the production of hydrogen peroxide by glucose oxidase in honeys. If the arbitrarily chosen water dilutions do not correspond to the appropriate water content and resulting viscosity, the results are uninterpretable and correlation creates false results.
2. The statement that “antibacterial activity of 12 selected honeys….. remained stable after complete protein digestion” is inadequately supported by results since the protein concentrations in honeys before and after proteinase K digestion have not been determined. To prove a complete protein digestion, the measurements of protein concentrations must be conducted.
3. Results of SDS-PAGE of honeys before and after proteinase K digestion presented in Fig. 6 are also questionable. Most of the honey proteins are glycosylated, including glucose oxidase. The most abundant protein group, MRJPs are high-mannose type glycoproteins that make them partially resistant to proteolytic degradation. Usually de-glycosylation is required prior to proteolytic hydrolysis and a combination of several proteases may be also necessary, in addition to proteinase K for complete protein digestion. There is lack of evidence that such methods were used in this study. It is therefore puzzling that not even traces of partially digested proteins are detectable in proteinase K- treated lanes presented in Fig. 6.
4. In contrast to the statement of line 296, “Currently, it is believed that H2O2 is solely produced by GOX-mediated conversion of glucose to gluconic acid under aerobic conditions in diluted honey” (White et al. 1963), the alternative, non-enzymatic production of hydrogen peroxide by polyphenol oxidation is well known for some time (Cao et al, 1997, Sakihama et al. 2002, Akagawa et al, 2003), including honey polyphenols (Brudzynski et al. 2012).
5. Line 298-302. There is apparent contradiction. MICs of honeys were found strongly dependent on hydrogen peroxide concentration (S. aureus (r = -0.51, P < 0.0001) and P. aeruginosa (r = -0.54, P < 0.0001). If GOX is completely removed by proteinase K digestion, then all observed hydrogen peroxide must be produced by some other pathways. Results reveal, at best, very modest correlations between total phenolic content and hydrogen peroxide concentrations (r = 0.32, P < 0.05). The massive production of H2O2 via polyphenol pathway cannot be substantiated.
Author Response
1. The statement that ”there is no relationship between the level of GOX and the level of H2O2 produced in similarly diluted blossom honeys” (Line 210) remains unsubstantiated. The authors have arbitrarily chosen 20% dilution, although there is lack of data in this manuscript how GOX activity is changing with honey dilutions. Moreover, accumulation of H2O2 is shown to be dependent on the amount of water and is changing with dilution in a non-linear manner approximating an inverted U-shape curve. In addition to that, this curve is different for each honey. The authors have not investigated this relationship and have again chosen arbitrarily 40% dilution without systematic study of H2O2 level in successive dilutions for different honeys. In such situation, changes of GOX level and changes of H2O2 concentration do not represent related changes and cannot be correlated at single selected points. The points of comparison are not corresponding to each other, so correlation cannot be found. The authors should include a systematic investigation of effects of honey dilutions on GOX activity on one hand, and hydrogen peroxide formation with dilutions on the other. Viscosity, sugar concentration and water content are the main factors influencing the production of hydrogen peroxide by glucose oxidase in honeys. If the arbitrarily chosen water dilutions do not correspond to the appropriate water content and resulting viscosity, the results are uninterpretable and correlation creates false results.
We understand your comments and suggestions. Based on other reviewer’s comments we decided to determine the hydrogen peroxide content in 100% honey. 40% honey solutions in buffer with pH 7.0 were prepared and used for immediate determination the content of hydrogen peroxide. Afterwards, the overall content of hydrogen peroxide was expressed as microgram of peroxide per g of honey. We again statistically analysed all correlations. See revised manuscript.
We assume that accumulated hydrogen peroxide during the honey ripening play some role in honey antibacterial activity which was determined in laboratory settings. To prove this theory, we determined MIC of artificial honey and artificial honey spiked with 50 ug/g of artificial honey. Artificial (synthetic) honey does not contain any proteins or polyphenols and was spiked only once with hydrogen peroxide. A significant decrease of MIC values was documented suggesting the initial concentration of hydrogen peroxide in honey is effective to inhibit bacterial growth.
As we mentioned in previous round of revision, we completely removed data regarding GOX activity.
2. The statement that “antibacterial activity of 12 selected honeys….. remained stable after complete protein digestion” is inadequately supported by results since the protein concentrations in honeys before and after proteinase K digestion have not been determined. To prove a complete protein digestion, the measurements of protein concentrations must be conducted.
We added a new evidence that support our statement. We detected GOX enzyme only in untreated honey samples. No visible monomeric form of GOX or any GOX fragments were detected in proteinase K-treated honey samples. Furthermore, we partially purified GOX enzyme from honey and used it for proteinase K treatment. We focused on enzymatic activity. After proteinase K - treatment of native partially purified GOX under the same conditions as honey treatment no activity was detected suggesting that proteinase K is able to digest GOX enzyme accompanying with loss of enzymatic activity within 30 min of incubation. This support data has not been shown in current study and will be used for further study dealing with the role of GOX in honey antibacterial activity.
3. Results of SDS-PAGE of honeys before and after proteinase K digestion presented in Fig. 6 are also questionable. Most of the honey proteins are glycosylated, including glucose oxidase. The most abundant protein group, MRJPs are high-mannose type glycoproteins that make them partially resistant to proteolytic degradation. Usually de-glycosylation is required prior to proteolytic hydrolysis and a combination of several proteases may be also necessary, in addition to proteinase K for complete protein digestion. There is lack of evidence that such methods were used in this study. It is therefore puzzling that not even traces of partially digested proteins are detectable in proteinase K- treated lanes presented in Fig. 6.
According to our knowledge and more than 20 years of experiences with honey and royal jelly proteins including MRJPs, GOX and Def-1, all honey proteins are sensitive for trypsin or pepsin digestion as well as proteinase K treatment. These observations have already been published in several papers. We added an immunoblot for clear evidence that GOX enzyme was digested and no partially digested GOX was detected in proteinase-K treated samples.
4. In contrast to the statement of line 296, “Currently, it is believed that H2O2 is solely produced by GOX-mediated conversion of glucose to gluconic acid under aerobic conditions in diluted honey” (White et al. 1963), the alternative, non-enzymatic production of hydrogen peroxide by polyphenol oxidation is well known for some time (Cao et al, 1997, Sakihama et al. 2002, Akagawa et al, 2003), including honey polyphenols(Brudzynski et al. 2012).
Thank you for your comment. We modified the statement and included the evidence of non-enzymatic production of hydrogen peroxide.
5. Line 298-302. There is apparent contradiction. MICs of honeys were found strongly dependent on hydrogen peroxide concentration (S. aureus (r = -0.51, P < 0.0001) and P. aeruginosa (r = -0.54, P < 0.0001). If GOX is completely removed by proteinase K digestion, then all observed hydrogen peroxide must be produced by some other pathways. Results reveal, at best, very modest correlations between total phenolic content and hydrogen peroxide concentrations (r = 0.32, P < 0.05). The massive production of H2O2 via polyphenol pathway cannot be substantiated.
Determination of hydrogen peroxide in honey solution after 24 h did not correspond properly with our antibacterial MIC assay. We focused on determination of original content of hydrogen peroxide in 100% honey. New obtained data from the hydrogen peroxide measurement in honey samples showed a strong correlation with polyphenol content (see revised manuscript).
Reviewer 3 Report
The introduced amendments and supplementary data significantly raised the scientific value of your article.
I recommend it for publication in Molecules.
Author Response
Thank you for your comments and improvements of our manuscript.
Round 3
Reviewer 1 Report
In the revised version of the manuscript, the authors made an effort to respond to the reviewer’s comments by adding two additional experiments. The reasons to conduct the additional experiments are not clear because the new results did not address the main concerns, that is, providing convincing results to support the concept that glucose oxidase is not responsible for production of H2O2 in honey. The reviewer’s suggestion to include quantitative evidence of complete removal of proteins after proteinase K treatment of honey (such as determination of protein concentration by Bradford assay) has not been implemented in the revised manuscript. This is a critical result to demonstrate that proteins, including GOX, are not participating in H2O2 production and antibacterial activity. Moreover, the results documenting the loss of GOX activity after proteinase K treatment are important for the study and should be presented (the authors stated “data not shown”, line 239), otherwise the manuscript is further raising objection as to the authors claim. Ultimately, the relationship or its lack between H2O2 levels and GOX levels has not been definitively established.
Under these circumstances, and since it is the third revision, I would leave the final decision about the suitability of the revision for publication up to the editor.
Concerns:
Some grammatical corrections and improvement in the sentence structure are needed.
Line 225-226. Please correct: The statistical analysis showed correlation between H202 and TP but not with the involvement of TP in its production. The latter was not investigated in this study.
Line 237-239: This sentence is not clear and should be rewritten and broken into two sentences.
Line 247- 248. This is an overstatement. The authors stated that “After confirming the proteinase-K effect on honey proteins”. The data presented in the manuscript are not sufficient to confirm the proteinase-K effect on honey proteins.
Line 248. Do the authors investigate the antibacterial activity of catalase? The sentence is not clear.
Line 250 -254: What was the purpose of these experiments?
Line 258-260. The sentence is too long and should be broken into two sentences.
Line 259. Unclear expression (“different changes”).
Line 332- 334: Sentence is unclear and had awkward and incorrect grammatical structure.
Line 339: Replace “among” with “between”.
Line 340: Change required. It should be, “Stronger correlation between MIC values against P. aeruginosa and H2O2 content than against S. aureus was found”.
Line 425: It should be “ripen honey” instead of “ripped honey”.
Author Response
Reply to the reviewer is in attached word file.
